# Oral Health Conditions and Physical Performance in Two Different Professional Team Sports in Germany: A Cross-Sectional Study

**DOI:** 10.3390/sports13070206

**Published:** 2025-06-25

**Authors:** René Schwesig, Paulina Born, Eduard Kurz, Stephan Schulze, Matti Panian, Robert Percy Marshall, Thomas Bartels, Andreas Wienke, Christian Ralf Gernhardt

**Affiliations:** 1Department of Orthopedic and Trauma Surgery, Martin-Luther-University Halle-Wittenberg, University Medicine, 06120 Halle, Germany; rene.schwesig@uk-halle.de (R.S.); eduard.kurz@uk-halle.de (E.K.); stephan.schulze@uk-halle.de (S.S.); matti.panian@uk-halle.de (M.P.); robert.marshall@redbulls.com (R.P.M.); 2University Outpatient Clinic for Conservative Dentistry and Periodontology, Martin-Luther-University Halle-Wittenberg, University Medicine, 06112 Halle, Germany; paulina.born5798@gmail.com; 3RasenBallsport Leipzig GmbH, Cottaweg 3, 04177 Leipzig, Germany; 4MVZ Sports Clinic Halle GmbH, Center of Joint Surgery, 06108 Halle, Germany; thomas.bartels@sportklinik-halle.de; 5Institute of Medical Epidemiology, Biostatistics and Informatics, Martin-Luther-University Halle-Wittenberg, University Medicine, 06112 Halle, Germany; andreas.wienke@uk-halle.de

**Keywords:** dental status, oral health, soccer, ice hockey, physical performance, posture, grip strength

## Abstract

Background: Oral health (OH) has been linked to overall health status and common physical performance (PP). Aim: The objective was to compare the OH and PP of two third league professional team players from different sports (soccer, ice hockey). Methods: Twenty-nine professional soccer players (mean ± standard deviation; age: 24.3 ± 4.1 years) and twenty-one ice hockey players (age: 27.7 ± 3.6 years) were investigated regarding several dental scores (DMFT, PSI, API, PBI) by one calibrated examiner. The PP diagnostic includes measurement of grip strength as well as postural stability and regulation using posturography. Results: Concerning dental scores, relevant sport-specific differences for PSI (1 vs. 2; *p* < 0.001), API (29 vs. 48; *p* = 0.001), and PBI (14 vs. 42; *p* < 0.001) in favor of soccer players were found. Ice hockey players reported significantly more tension of the temporomandibular joint (*p* = 0.004) and showed lower readiness for regular check-ups (*p* < 0.001) and additional oral hygiene (*p* = 0.045). In contrast, ice hockey players were more balanced and displayed a higher level of weight distribution (*p* < 0.001), especially in the anterior-posterior direction (*p* = 0.002). Conclusions: Based on more intensive oral hygiene and dental care, the investigated professional soccer players showed partially better OH conditions compared to the ice hockey players examined. In summary, the OH and PP results reflect the different sport-specific requirements and related training impacts on athletes.

## 1. Introduction

Oral health is an integral component of general health and well-being [1,2]. However, in the context of professional sports, it remains an often neglected and underestimated aspect, despite mounting evidence of its clinical relevance [3,4]. Currently, there is no study which addresses the potential links among oral health, muscle function, and postural control in a sportive setting, especially in team sports. Existing studies have only been carried out in a clinical setting [5,6] or addressed elderly people [7,8].

Only a few studies have indicated a considerable prevalence of oral diseases and injuries among elite athletes, including dental caries, periodontal disease, and so-called non-carious lesions like dental erosion [9,10]. For example, in ice hockey, the oral region was the third most common body part (16% of total injuries) to be injured, after the arms and legs [11]. Based on an analyzed 15-year period in Alberta (Canada), the authors suggested preventing oral and dental injuries through encouraging the use of mouthguards and full-face protection [11]. However, the associations among oral health, muscle function, and postural control remain unclear.

These facts and conditions not only reflect suboptimal oral hygiene, but may also exert a measurable impact on athletic performance and quality of life [12,13,14]. In a cross-sectional study by Gallagher et al. [13], 50% of athletes were found to have carious lesions, while 77% exhibited gingival bleeding upon probing, and 82% reported that their oral health influenced their athletic performance. Similarly, Needleman et al. [12] observed that over half of the Olympic athletes they examined presented with caries, 45% with erosion, and 75% with gingivitis. Although only 18% explicitly noted a negative influence on performance, the high prevalence of oral pathologies suggests a substantial need for preventive and therapeutic dental care. Opposite findings were reported by Kragt et al. [15], where a low participation rate (15%) among invited Dutch athletes may have introduced significant bias. Nevertheless, even within that limited sample, 43% of participants exhibited dental treatment needs. Konviser et al. [16] identified a high prevalence of oral disease among soccer players in England (age: 16–18 years), with over half presenting with untreated dental caries, 77% showing signs of gingivitis, and notable levels of tooth wear and dental trauma. Although ice hockey is significantly more prone to acute dental trauma than other sports, no studies of dental health in professional ice hockey have been identified. The only publication we are aware of describes poor dental hygiene in more than 50% of adolescent athletes [17]. This suggests that the incidence in adults is likely to be correspondingly higher. Poor oral hygiene practices, infrequent dental attendance, and high consumption of sugary and acidic drinks were common, highlighting the urgent need for integrated oral health promotion within youth soccer development programs [18]. It should be noted that chewing tobacco and snus are often used in professional sports, which may contribute to further deterioration of oral health [19,20]. In addition, high-intensity athletic training itself is a risk factor for colonization with pathogenic bacteria due to suppression of the oral immune system, which may contribute to an increased incidence of poor oral hygiene in competitive athletes [21].

Beyond the scope of athletic performance, poor oral health has been implicated in the pathogenesis of systemic diseases [22,23,24,25,26,27]. Evidence from longitudinal and observational studies has established associations among socio-economic factors, periodontal disease, and an elevated risk of cardiovascular events, including coronary heart disease and stroke [28,29,30]. Furthermore, excessive tooth loss has been linked to increased mortality from both cardiovascular and gastrointestinal causes [31]. There is a growing understanding of the influence of the oral microbiome on the whole microbiome and the influence of inflammatory activity in the oral cavity on general inflammation in the body [32,33,34,35].

Regarding these influencing factors, it is reasonable to hypothesize that professional ice hockey and soccer players, due to the sport’s high physical intensity and the increased risk of injuries and dental trauma, may be particularly susceptible to poor oral health and its associated consequences. Furthermore, dental lesions and periodontal disease are well known and widespread clinical findings in professional athletes. Understanding the prevalence of oral diseases in this population, as well as the potential implications for both performance and systemic health, is essential for developing targeted preventive strategies and improving the overall health outcomes of athletes.

Therefore, the aims of this cross-sectional study were to assess oral hygiene and dental health in professional soccer and ice hockey players, to compare the two team sports, and to estimate the association regarding basic physical performance (strength and posture). For the null hypothesis, it was assumed that there were no differences in both dimensions between soccer and ice hockey players in local third league professional teams from Germany.

## 2. Materials and Methods

### 2.1. General Study Design

A cross-sectional research design was used to explore and clarify dental status and physical performance in two different professional team sports (Figure 1). As a basic part of the data collection, anthropometric measurements were captured using a weight scale and stadiometer (for body mass and height). The physical performance diagnostic (grip strength, postural control, sport specific dimensions) was conducted at the start of the pre-season (soccer: June; ice hockey: August) in a defined order after the dental investigation.

The study protocol was approved by the local Ethics Committee of the Martin Luther University Halle-Wittenberg (reference number: 2022-011) and conformed to the Declaration of Helsinki [36]. Screened participants were extensively informed and additionally received written information about the entire study in advance and signed consent forms indicating their willingness to be part of the investigation. For players younger than 18.0 years (n = 3; age range: 17.2–42.1 years), parental/guardian consent was obtained.

The dental investigation was conducted by one calibrated and experienced dentist, to avoid interobserver differences. The same aspect was considered in the physical performance diagnostic process, which was carried out by an experienced and calibrated sports scientist. Athletes were initially instructed and trained to allow performing the tests as accurately as possible. Furthermore, they were motivated to give maximal effort with strong verbal encouragement during testing of grip strength. The tests were conducted in an air-conditioned research laboratory under the supervision of experienced investigators, at the same time of day (9 to 12 am) and with similar environmental conditions (~20 °C and ~60% humidity).

### 2.2. Participants

Fifty male professional soccer and ice hockey players (Table 1) were included in this study. All athletes were members of a third league ice hockey team (n = 21) or a third league soccer team (n = 29). The weekly workload was 5–6 workouts (duration per workout: 60–120 min) and one (soccer) or two (ice hockey) matches per regular week during the season, which underlines the professionalism of the players. The workload is comparable to the second or first leagues.

Based on the professional history of all players, all subjects were highly familiarized with the performance diagnostic approach.

Days on which players submitted an illness certificate were counted as sick and injured days. Therefore, the number of injuries did not depend subjectively on the players, but objectively on the team doctor issuing a certificate and its duration.

### 2.3. Testing Procedures

#### 2.3.1. Questionnaire for Oral Health

First, dental anamnesis (Figure 1) using a standardized questionnaire was conducted to describe the status of oral health and hygiene. Most of the questions were structured dichotomously (0 = no, 1 = yes). We used the following questions:Did you have any injuries in the past two years?Did you ever have any surgery?Did you have any traumatic injuries related to sport activities inside your oral cavity?Do you have toothache?Do you suffer from tensions around your jaw joint/shoulder/neck?Do you use any additional oral care products?

#### 2.3.2. Dental Indices and Scores

The *DMFT (Decayed, Missing, and Filled Teeth)* index is a common index used in dentistry to assess the level of dental decay and health of individuals on a tooth level. The oral health of the athletes was evaluated during clinical examinations using a flat surface mirror, exploration and periodontal probes, gauze, and compressed air under artificial lighting conditions. The DMFT index, proposed by the World Health Organization (WHO), was calculated, referring to the number of decayed, missing, and filled teeth in the permanent dentition. It counts the number of teeth that are decayed (D), missing due to decay (M), or filled because of decay (F), providing a total score that reflects a person’s experience with dental decay. This index helps in evaluating oral health status and planning public health interventions [37].

The *PSI (Periodontal Screening Index)* is a common and standardized method used by dentists to assess the periodontal health of a patient’s gums and detect signs of periodontal disease in five different degrees. The mouth is divided into sextants (S1-S6), and each is scored based on pocket depth, overhanging filling edges, bleeding, and the presence of tartar or recessions. Each tooth (excluding the wisdom teeth) is then examined using a millimeter-scale periodontal probe. The periodontal probe is moved circularly through the pockets and used to probe at six points: mesiobuccal, buccal, distobuccal, mesiooral, oral, and distooral. The possible findings are summarized per sextant with codes 0 to 4. The highest value in the sextant is decisive. This helps identify areas needing further examination or treatment [38].

The *API (Approximal Plaque Index)* measures the presence of dental plaque on the approximal (interdental) surfaces of teeth. After applying a disclosing agent, the dentist examines specific tooth surfaces for plaque accumulation. The percentage of surfaces with plaque indicates the patient’s oral hygiene status: an API over 35% suggests insufficient hygiene, while 25% or lower reflects very good oral hygiene [38].

The *PBI (Papillary Bleeding Index)* uses the same probing technique as the modified Sulcus Bleeding Index (SBI) but is measured on opposite aspects of the dentition (on the oral side in the first and third quadrants and on the buccal side in the second and fourth quadrants). Unlike the SBI, the PBI not only detects the presence of interdental bleeding, but also quantifies its severity on a four-point scale: grade 1 indicates a single bleeding point; grade 2 denotes a fine line of blood or multiple bleeding points; grade 3 means the entire interdental papilla fills with blood; and grade 4 represents profuse bleeding with an immediate droplet flowing over the tooth and gingiva. The overall PBI score is calculated by dividing the sum of all recorded bleeding grades by the total number of interdental sites examined [39].

#### 2.3.3. Dynamometry-Grip Strength

Data collection regarding grip strength was based on ASHT (American Society of Hand Therapists) guidelines and performed using a hand dynamometer (SH5001 SAEHAN, Changwon-si, Republic of Korea). The test procedure following the ASHT guidelines includes several important aspects (Figure 2a,b) [40]:-Participant seated in an upright posture with both hips and knees in 90° flexion with feet flat on the floor-The extremity itself has no contact with the body-The elbow is flexed at 90°, forearm in neutral position, wrist slightly extended (0° to 30°), and ulnar deviation ranging from 0° to 15°.-The non-dominant extremity should be relaxed at the side simultaneously.

The measurements were conducted with and without jaw clenching in order to judge the influence of maximal intercuspal position of the mandibula in dental occlusion regarding upper body strength and postural control [41,42]. According to different sport-specific requirements (soccer vs. ice hockey), and to provide a more valid comparison between team sports, the values from the left and right side were added (absolute combined grip strength) and related to the body mass (relative combined grip strength; kg/kg bm).

Every athlete performed two attempts (30 s rest between the repetitions) per side and test situation. For statistical analysis, the largest value per side was used. The entire approach (calculation of combined grip strength and use of maximum values) was based on Quinney et al. [43].

#### 2.3.4. Dynamometry-Posturography (Posture Stability and Regulation)

Posture was measured with a posturographic system (Interactive Balance System IBS, neurodata Gmbh, Vienna, Austria) in order to assess postural stability, regulation, and weight distribution. Four independent force plates form the basis for this dynamometric assessment (sampling rate: 32 Hz). Fast Fourier transformation (FFT) is the crucial feature for the IBS to transform the force-time-signal in a spectrogram including sway intensities (different frequency ranges) and amplitudes. On this basis, it is possible to assign different functional frequency bands (F1, F2–4, F5–6, F7–8) to several postural subsystems (F1 = visual and nigrostriatal system; F2–4 = peripheral-vestibular system; F5–6 = somatosensory system; F7–8 = cerebellar system) [44,45]. In contrast to such process parameters, it is also possible to measure on the “product level”. Such motor output parameters can be defined as stability indicator (ST = postural stability), synchronization (synch = foot coordination), weight distribution index (WDI), forefoot–hindfoot ratio (heel), and left–right ratio (left) [46]. All players performed eight single trials (32 s, without shoes) under different test conditions with respect to head and neck position and visual and somatosensory input (Figure 3).

The included athletes were instructed as follows: stand upright, with weight evenly balanced on the two force plates and eyes focused straight forward on an individually adjusted target related to the body height. The validity [47] and intraobserver reliability [48] of the IBS are often reported. An additional advantage of this tool is the possibility of comparing the data with gender- and age-specific reference data from a large (n = 1724) population-based asymptomatic sample over the whole life span [49].

### 2.4. Statistical Analysis

All statistical analyses were performed using SPSS version 28.0 for Windows (IBM, Armonk, NY, USA). Descriptive statistics (mean, standard deviation, 95% confidence interval (95% CI)) were reported for all metric scaled parameters. For the ordinal scaled parameters (e.g., dental scores), medians and percentiles (e.g., interdecile range) were presented. All variables were examined for normal distribution (Shapiro–Wilk Test).

Mean differences between team sports (soccer vs. ice hockey) were tested using a univariate general linear model. Initially, the relevance level was defined as partial eta-squared (η_p_^2^) > 0.15 [50].

The effect size d (the mean difference between scores divided by the pooled SD) was also calculated for all parameters [51].

To evaluate d or η_p_^2^, the following recommendations were used: d ≥ 0.2, d ≥ 0.5, d ≥ 0.8, or η_p_^2^ ≥ 0.01, η_p_^2^ ≥ 0.06, η_p_^2^ ≥ 0.14, indicating small, medium, or large effects, respectively [52].

Additionally, a reference database [49] was used to select the suitable age range (20 to 30 years) as a comparison sample. For better interpretation of the revealed posturographic data, we reported the 50th percentile (median) and the interquartile ranges (25th and 75th percentile).

Associations between metric variables were analyzed using Pearson’s product-moment correlation (r) and interpreted as negligible (<0.1), weak (0.1–0.4), moderate (0.4–0.7), strong (0.7–0.9), or very strong (>0.9) [52]. A r^2^ value > 0.7 (explained variance > 50%) was considered relevant and reported. For ordinal scaled parameters, Spearman product-moment correlation was used.

Frequency distributions from dichotomous variables were compared using the chi-squared test.

Binary logistic regression (univariate and multivariate analysis) was performed (method: inclusion) to estimate the interaction between metric physical performance parameters (posture, grip strength) and dichotomous (1 = yes, 0 = no) oral health parameters (e.g., tension of the temporomandibular joint). Odds ratios (ORs) including 95% CI, *p*-value, and Nagelkerkes r^2^ as an estimator for the whole multivariable model were described.

A priori power calculation was performed to determine the appropriate sample size (primary outcome: DMFT). The sample size calculation for the Mann–Whitney U test was performed according to Noether [53], based on the following assumptions: a two-sided hypothesis, alpha level: *p* < 0.05, power 1-β: 0.8, mean (SD) of group 1: 2 (1), and mean (SD) of group 2: 3 (1). The analysis revealed that 20 participants per group were necessary to detect relevant differences.

## 3. Results

### 3.1. Characteristics of the Included Participants

In total, 50 professional athletes (29 soccer, 21 ice hockey players) were included in the present study. The soccer players are younger and showed a markedly lower BMI than the ice hockey players (Table 1).

### 3.2. Normal Distribution

The following metric posturographic variables did not display a normal distribution: F1 (*p* < 0.001), F2–4 (*p* = 0.028), F5–6 (*p* = 0.001), F7–8 (*p* < 0.001), ST (*p* < 0.001), and WDI (*p* < 0.001).

### 3.3. Results of the Dental Examination

The dental investigation provided some differences between the soccer and ice hockey players (Table 2). Apart from DMFT, the ice hockey players showed a lower level of oral health, as evaluated by the scores PSI, API, and PBI.

Concerning several oral health conditions depending on team sport (ice hockey vs. soccer), the following aspects were examined using a questionnaire:previous illnesses (5% vs. 11%, *p* = 0.480),medication intake (5% vs. 4%, *p* = 0.807),number of sick days (3.5 ± 2.1 vs. 3.3 ± 3.9, *p* = 0.797),number of injuries (80% vs. 68%, *p* = 0.351),past operations (70% vs. 50%, *p* = 0.166),traumatic injuries in the mouth (30% vs. 21%, *p* = 0.499),tooth pain (10% vs. 7%, *p* = 0.724),bleeding gums (30% vs. 18%, *p* = 0.327),grinding teeth (0% vs. 14%, *p* = 0.077),tension of the temporomandibular joint (35% vs. 4%, *p* = 0.004),previously treated orthodontically (35% vs. 46%, *p* = 0.428),regular check-ups (40% vs. 4%, *p* < 0.001),number of times teeth brushed per day (two times daily: 85% vs. 93%, *p* = 0.227),additional oral hygiene procedures (35% vs. 64%, *p* = 0.045),satisfaction with oral hygiene (80% vs. 89%, *p* = 0.369),changes in teeth (40% vs. 43%, *p* = 0.843),impact on competitive sport (5% vs. 14%, *p* = 0.299).

Regarding tension of the temporomandibular joint, a significant difference between soccer and ice hockey players was calculated. In total, 7 of 20 (35%) ice hockey players reported such problems (soccer: 1/28 (4%)). The largest difference between sports was observed for regular check-ups. Overall, 96% (27/28) of soccer players used this opportunity (ice hockey: 8/20 (40%). Furthermore, soccer players (64%, 18/28) showed a higher readiness for additional oral hygiene procedures (i.e., dental floss) than ice hockey players (35%, 7/20).

### 3.4. Comparison Ice Hockey vs. Soccer and Comparison Based on Specific Reference Data

Regarding grip strength under different measurement conditions, we could not detect any relevant differences (Table 3). Considered absolutely, the ice hockey players consistently showed a slightly higher strength level compared with the soccer players. In relation to body mass, the soccer players revealed under both conditions higher values.

The analysis regarding posturographic parameters (Table 4) contains a comparison using reference data based on comprehensive data collection from asymptomatic subjects [49] (men, aged 20–30 years, n = 277).

Apart from frequency bands 5–6 (somatosensory system) and 7–8 (cerebellar system) and the stability indicator (postural stability), ice hockey players showed consistently higher postural performance level than soccer players. Only for WDI and heel (anterior-posterior weight distribution) were relevant differences observed (Table 3).

Compared to the asymptomatic reference data, soccer players showed a slightly higher level in F5–6 (somatosensory subsystem), whereas the ice hockey players displayed slight superiority in the following three parameters: WDI, heel, synchronization (Table 4).

### 3.5. Associations Between Oral Health and Performance Parameters

Correlations of practical value (r > 0.5) could not be found between oral health and performance parameters. The largest, but not relevant, relationship was detected for DMFT and F1 (r = −0.355).

Prior to the final logistic regression analysis (Table 5), the independent variables were checked regarding their influence on the numeric stability of the statistical model because of the limited sample size. After this check, three variables (grip strength, F5–6, F7–8) had to be removed in order to avoid numeric instability of the model.

Only for mouth injuries, posturographic variables (F1, F2–4, synchronization) have a relevant predictive power (explained variance: 53%).

In a second step, we separately analyzed the appropriate posturographic data regarding the criterion “movement of the neck during the measurements”. In this context, we calculated the difference for the unilateral relevant (based on the first model) parameters (F1, F2–4, ST) between the last four test positions (HR, HL, HB, HF) with neck movements and the first four test positions (NO, NC, PO, PC) without any neck movements. The additional inclusion of these three parameters led to a large increase in the explained variance for all oral health variables by the binary logistic model (Table 6).

## 4. Discussion

This is one of the first studies comparing professional soccer and ice hockey players in terms of oral health, oral hygiene, and several dimensions of physical performance (strength and posture) in sufficiently large numbers (n = 50). This is against a background of continued underestimation of the impact of oral health on the physical performance of professional athletes.

The results of this study show that soccer players practice better oral hygiene and therefore have better oral health, as measured by dental scores (PSI, API, PBI). Our hypothesis that there were no differences in strength, as well as postural stability and regulation, between soccer and ice hockey was partly confirmed. Regarding grip strength under different conditions (in maximal intercuspal or physiological rest position of the mandibula), we did not find any relevant differences. In contrast, ice hockey players were more balanced, especially in the anterior-posterior direction, than soccer players. Compared with asymptomatic subjects, soccer players displayed a small advantage in the somatosensory subsystem, and ice hockey players had an advantage concerning weight distribution (WDI, heel = forefoot-hindfoot ratio) and foot coordination (synchronization).

The oral health results showed clear deficits, particularly in the case of ice hockey players. The high DMFT values for ice hockey players (median: 5.5) indicate a considerable caries burden. The oral hygiene parameters (API, PBI) and the periodontal values (PSI) showed potential for improvement, especially for ice hockey players. The results of the present study support findings from other authors [13,54,55,56], particularly with regard to caries prevalence and periodontal status.

Gay-Escoda et al. [54] examined a comparable sample of 30 professional soccer players (mean age: 21 ± 1.6 vs. 24 ± 4.1 years in the present study). The authors reported a clearly higher mean DMFT of 5.7 in comparison to our investigated soccer players (median: 3.5). Chantaramanee et al. [55] found that the oral health of soccer players (n = 25, mean age: 27.5 ± 4.72 years) was very poor, connected with a high percentage of dental caries. Gallagher et al. [13] provided some similarities with regard to oral health, especially concerning caries prevalence (measured via the DMFT index). The authors found carious lesions in 49% of athletes, which corresponded to an average of two decayed teeth per athlete. Soccer and rugby players were most frequently affected [13]. In comparison, the average DMFT value of 5.5 in the present study is higher, which indicates a higher caries burden in the examined ice hockey sample.

Comparable results can also be seen when looking at periodontal status. Gallagher et al. [13] and the present study point to a high prevalence of gum disease, as measured using the Periodontal Screening Index (PSI) and the Basic Periodontal Examination (BPE). Both studies confirmed the correlation between periodontal disease and systemic inflammatory markers.

Gallagher et al. [13] found that, in terms of self-assessment of oral hygiene, 69% of athletes rated their oral hygiene as good or very good. Eighty-five percent of athletes in the present study stated that they were satisfied with their oral hygiene. Gallagher et al. [13] reported that 32% of participants reported that their oral health negatively affected their athletic performance. This number was significantly lower in the present study: only 8% and 10% of soccer and ice hockey players, respectively, reported toothache and expressed the opinion that oral health affected their athletic performance negatively. Similar results were found by Needleman et al. [56], who also focused on team sports. Both studies showed comparable DMFT values. While Needleman et al. [56] determined an average DMFT index of 4.6, the present study revealed a slightly higher average value of 5.1, which suggests a higher caries burden among the athletes examined.

There are also parallels in periodontal status: Needleman et al. [57] found that 80% of the players suffered from gingivitis. This is consistent with the results of the present study, which also indicate the frequent occurrence of gingivitis, with an average PSI of 1.8.

Sevindik et al. [17] also detected lower oral health for ice hockey players. Compared with other winter sports (alpine discipline skiing, snowboarding, biathlon, ski jumping), ice hockey players (n = 22, age: 15.1 ± 0.4 years) were the most affected by untreated dental caries (27%). Regarding DMFT, young ice hockey players displayed similar results (5.9) to the players we investigated (5.5).

Grip strength is a simple, valid, and reliable parameter for estimating overall muscular strength, a key component of health-related fitness and physical performance [58]. Measurement of grip strength under maximum intercuspation in ice hockey players opens up interesting perspectives for research into neuromuscular relationships. In this way, it was possible to investigate the extent to which the jaw muscles play a role in general strength development.

Toong et al. [58] investigated 690 male and female youth ice hockey players (age range: 10–16 years). Based on these reference data, the grip strength data of our sample moved between percentile 75 (89 kg) and percentile 90 (111 kg). Gouveia et al. [59] investigated 83 professional male soccer players (age range: 17–30 years) regarding different dimensions (grip strength, vertical jumping, isokinetic strength). The measured grip strength was clearly lower (range depending on age: 78.8–96.9) than in our study (101 or 106 kg). Hümer et al. [60] collected data from 200,389 asymptomatic subjects. According to these data (age range: 19–29 years), our current investigated group was around percentile 75 (110 kg).

A relevant limitation of this investigation is the inconsistent recording of sick days by different team doctors. Additionally, the team sport-specific differences in the duration of seasons and the number of matches should be taken into consideration. Furthermore, the different sport-specific demands require an adapted performance diagnosis, which is then no longer comparable between sports. For example, we conducted the endurance performance diagnostic on a treadmill for soccer players and on a bike for ice hockey players. Furthermore, we performed an ice hockey-specific complex test [61]. Therefore, in this work we have limited ourselves to comparison of oral health data with general performance diagnostics (grip strength, posture). In the next step, we will published the analysis of sport-specific data related to oral health in two separate publications.

Finally, the validity and external generalizability of the study’s findings are limited to the third league for soccer and ice hockey. Consequently, investigation of second and first league teams should be addressed in future studies in order to elevate the range of validity.

## 5. Conclusions

In conclusion, oral health and readiness for additional oral hygiene are more pronounced for soccer players than ice hockey players, with corresponding consequences for oral health. Special demands regarding posture in ice hockey are responsible for a markedly more balanced weight distribution. Based on missing interactions between different dimensions, it seems that posture, grip strength, and oral health are not relevantly related to each other.

## Figures and Tables

**Figure 1 sports-13-00206-f001:**
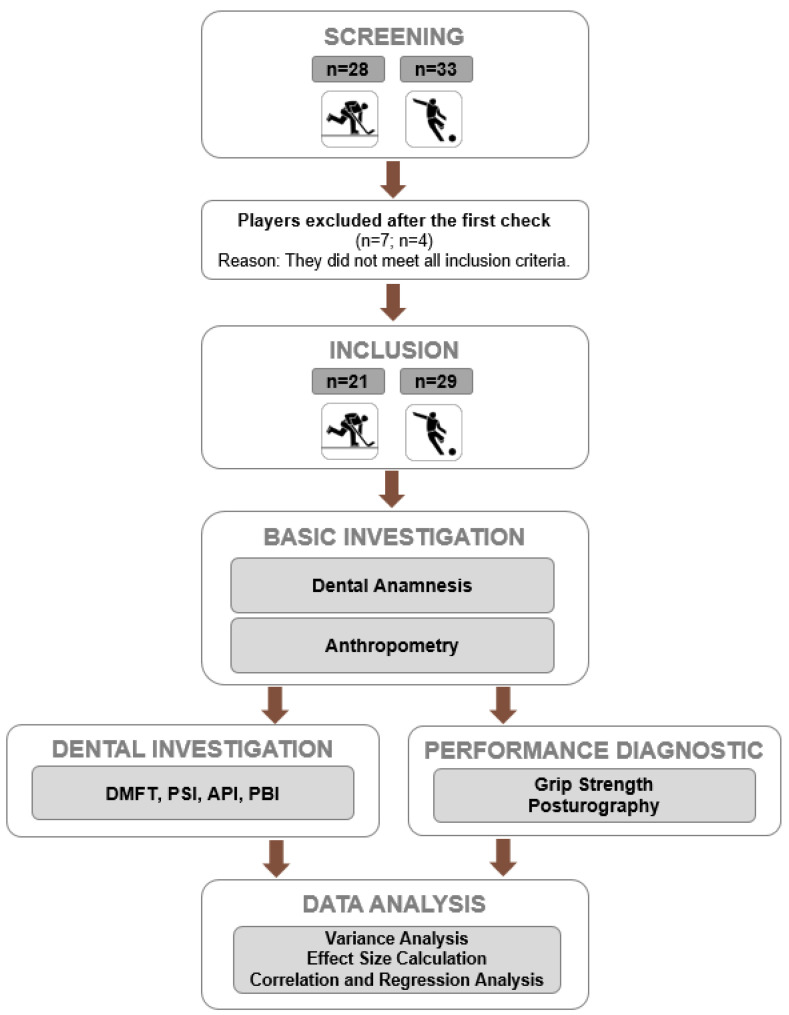
Flow diagram of the study design.

**Figure 2 sports-13-00206-f002:**
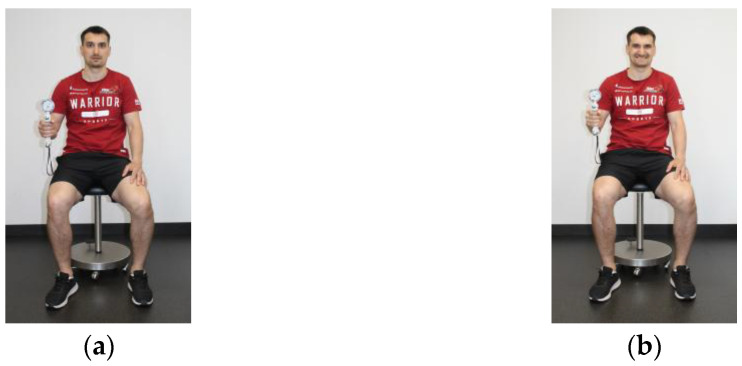
(**a**,**b**) Measurement of grip strength based on ASHT guidelines using the Hydraulic Hand Dynamometer SH5001 (**a**) in physiological rest position of the mandibula and (**b**) in maximal intercuspal position of the mandibula.

**Figure 3 sports-13-00206-f003:**
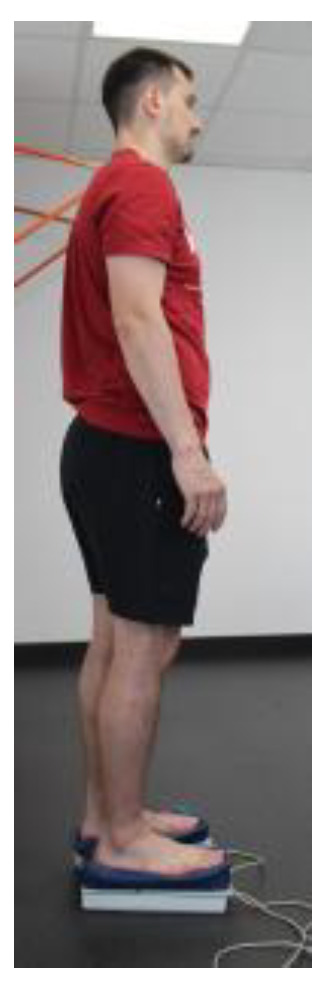
Measurement of postural stability and regulation using the IBS. The first test position (NO = normal head position, eyes open, without foam pads) is shown.

**Table 1 sports-13-00206-t001:** Demographic and anthropometric characteristics of ice hockey (n = 21) and soccer players (n = 29). Data are mean ± standard deviation (SD). Relevant differences and maxima are marked in bold.

	Soccer	Ice Hockey	Variance Analysis	
			*p*	η_p_^2^
Age [years]	24.3 ± 4.13	**27.7** ± 3.55	**0.004**	**0.16**
Height [m]	**1.83 ± 0.06**	1.82 ± 0.07	0.555	0.01
Body mass [kg]	80.3 ± 7.59	**86.5** ± 9.10	0.011	0.13
BMI [kg/m^2^]	24.0 ± 1.78	**26.1** ± 1.82	**<0.001**	**0.27**
Sick days [d]	3.25 ± 3.93	**3.50** ± 2.09	0.797	0

**Table 2 sports-13-00206-t002:** Comparison of dental scores depending on team sport. Values are given as median (interdecile range).

Parameter	Soccer (n = 28)	Ice Hockey (n = 20)	Mann–Whitney U Test (*p*)
DMFT	3.5 (0–9.2)	5.5 (0.2–13.0)	0.062
DT	0 (0–1.1)	0 (0–2.0)	0.969
MT	0 (0–2.0)	0 (0–2.0)	0.105
FT	3.0 (0–7.2)	4.0 (0.1–11.0)	0.152
PSI	1.0 (0.9–2.1)	2.0 (2.0–3.0)	<0.001
API (%)	28.5 (9.4–54.3)	47.5 (22.3–72.7)	0.001
PBI (%)	14.0 (0–39.3)	41.5 (11.3–67.3)	<0.001

**Table 3 sports-13-00206-t003:** Comparison of combined (left + right) grip strength depending on team sport and jaw clenching. Results are reported as mean ± SD (95% CI). Significant relevant differences (criteria: *p* < 0.05 and η_p_^2^ > 0.15 and d > 0.8) and performance maxima are marked in bold.

Parameter	Soccer (n = 29)	Ice Hockey (n = 21)	Variance Analysis		
			*p*	η_p_^2^	d
**Combined grip strength in physiological rest position of the mandibula**					
absolute (kg)	101 ± 11.3 (96.2–105)	**106** ± 13.0 (100–111)	0.151	0.04	0.41
relative (kg/kg BM)	**1.26** ± 0.16 (1.20–1.33)	1.23 ± 0.17 (1.16–1.30)	0.493	0.01	0.42
**Combined grip strength in maximal intercuspal position of the mandibula**					
absolute (kg)	106 ± 13.4 (101–112)	**110** ± 13.4 (104–116)	0.383	0.02	0.30
relative (kg/kg BM)	**1.33** ± 0.17 (1.26–1.39)	**1.28** ± 0.17 (1.20–1.35)	0.313	0.02	0.29

**Table 4 sports-13-00206-t004:** Comparison of posturographic parameters depending on team sport and reference data. Results are reported as mean ± SD (95% CI). Reference data: n = 277; age range: 20–30 years. Significant relevant differences (criteria: *p* < 0.05 and η_p_^2^ > 0.15 and d > 0.8) and performance maxima are marked in bold.

Parameter	Soccer (n = 27)	Ice Hockey (n = 21)	Reference Data	
			Mean (SD)	95% CI
F1	19.3 ± 6.09 (16.9–21.7)	**17.5** ± 6.17 (14.8–20.1)	14.6 (4.23)	14.1–15.2
p/η_p_^2^/d	0.319/0.02/0.29			
F2–4	8.76 ± 1.95 (8.03–9.50)	**8.68** ± 1.81 (7.85–9.51)	8.15 (1.94)	7.87–8.43
p/η_p_^2^/d	0.884/0/0.04			
F5–6	**3.55** ± 0.82 (3.22–3.89)	3.69 ± 0.94 (3.31–4.07)	3.58 (0.90)	3.45–3.70
p/η_p_^2^/d	0.587/0.01/0.16			
F7–8	**0.70** ± 0.17 (0.63–0.78)	0.71 ± 0.22 (0.62–0.79)	0.66 (0.19)	0.64–0.69
p/η_p_^2^/d	0.935/0/0.05			
ST	**20.1** ± 4.98 (17.9–22.3)	20.9 ± 6.36 (18.4–23.3)	16.7 (3.86)	16.2–17.3
p/η_p_^2^/d	0.637/0.01/0.14			
WDI	6.59 ± 2.74 (5.73–7.45)	**4.27** ± 1.25 (3.29–5.24)	5.21 (2.17)	4.95–5.47
p/η_p_^2^/d	**<0.001/0.22/1.16**			
Heel (%)	42.0 ± 9.43 (38.8–45.1)	**49.6** ± 6.04 (46.1–53.2)	46.6 (8.07)	45.6–47.6
p/η_p_^2^/d	**0.002/0.19/0.98**			
Left (%)	50.6 ± 2.84 (49.3–51.8)	**49.7** ± 3.78 (48.3–51.1)	50.1 (3.29)	49.7–50.5
p/η_p_^2^/d	0.370/0.02/0.27			
Synchronization	603 ± 133 (549–657)	**670** ± 149 (608–731)	625 (119)	608–642
p/η_p_^2^/d	0.108/0.06/0.48			

**Table 5 sports-13-00206-t005:** Summary of the logistic regression (multivariable analysis) for physical performance parameters (independent variables) depending on selected oral health parameters (dependent variables). Nagelkerkes r^2^ > 0.5 (explained variance > 50%) is marked in bold. Synch. = synchronization. Please note that the different dependent variables were separately used in the model.

Dependent Variables	Independent Posturographic Variables				
	r^2^	Parameter	OR	95% CI	*p*
general injuries	0.10	-			
general surgeries	0.11	-			
traumatic mouth injury	**0.53**	F 1	0.75	0.60–0.93	0.01
		F 2–4	3.88	1.33–11.4	0.01
		Synch.	1.01	1.00–1.02	0.02
tooth pain	0.05	-			
tension of the temporomandibular joint	0.39	-			

**Table 6 sports-13-00206-t006:** Summary of the logistic regression (multivariable analysis) for physical performance parameters depending on selected oral health parameters. Nagelkerkes r^2^ > 0.5 (explained variance > 50%) is marked in bold. Please note (in line with Table 5) that the different dependent variables were separately used in the model.

Dependent Variables	Independent Posturographic Variables				
	r^2^	Parameter	OR	95% CI	*p*
general injuries	0.20	-			
general surgeries	0.37	-			
traumatic mouth injury	**0.58**	F 1	0.75	0.59–0.95	0.02
		F 2–4	4.59	1.28–16.5	0.02
		Synch.	1.01	1.00–1.02	0.03
tooth pain	0.10	-			
tension of the temporomandibular joint	**0.52**	WDI	0.28	0.09–0.95	0.04

## Data Availability

The data are available upon request from the first or corresponding author.

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
