# Peer review of "Oral Health Conditions and Physical Performance in Two Different Professional Team Sports in Germany: A Cross-Sectional Study"

_sports, 2025, doi:10.3390/sports13070206_

Round 1
Reviewer 1 Report
Comments and Suggestions for Authors
The manuscript addresses an interesting topic at the intersection of sports medicine and dental health, offering new insights by comparing oral health and physical performance parameters in athletes from two different team sports. The inclusion of both dental indices and physical performance measures adds particular value and further enriches the manuscript. However, although the study is interesting and potentially significant, the manuscript requires revision and additional clarifications. Therefore, I recommend major revisions before the manuscript can be considered for publication.
In the introduction section of the manuscript, the prevalence of oral diseases and injuries among professional athletes is discussed in detail, with particular emphasis on dental trauma and caries. However, although the study’s objective clearly states the intention to assess the relationship between oral health and basic physical abilities—specifically strength and posture—these aspects are not mentioned in the introduction. Given that strength and postural stability are key variables in this research, I recommend that the introduction be expanded to include a brief review of the literature, or at least a short discussion on the potential link between oral health, muscle function, and postural control. This would provide a more coherent logical connection between the introduction, the study aim, and the overall research design.
The study included 50 professional athletes from third league teams (29 soccer and 21 ice hockey players), yet the manuscript does not sufficiently justify the sampling method or explain how this sample size was determined. It remains unclear whether this number of participants is based on a formal power calculation, convenience sampling, or team availability. Moreover, while these athletes are described as professionals, they represent third league teams—raising the question of how representative they are of broader professional populations in their respective sports.
Given that the study aims to assess both oral health and physical performance parameters and compare between two sports, a more detailed justification of sample adequacy is warranted. I recommend the authors clearly explain: how the sample size was calculated (e.g., was a priori power analysis conducted?), why third league players were selected, and whether this cohort can be generalized to other levels of professional sport. This clarification is important for assessing the validity and external generalizability of the study’s findings.
Reviewer 2 Report
Comments and Suggestions for Authors
This is an interesting and well written manuscript, which has been well presented.
Page 3, Line 96: I feel that this statement is really a 'null hypothesis' rather than a hypothesis, so a small alteration should be made to this.
Page 6, Line 235: Is absolute grip strength related to body mass? If this is the case a short explanation and reference to this is needed. If this is your own interpretation then please explain why you have used this approach Why did you use the largest value per side rather than a mean value overall? Please explain.
Page 7, Line 308: Whilst conforming to the referencing style of the journal, it would be helpful to have the date of the reference in the text. For example Noether.
When asking about injuries it is unclear whether the question is directed to oro-facial injuries or injuries in general. I noted that the ice-hockey players reported less injuries; could this be because the question is rather subjective and ice-hockey players may not consider some minor knocks as injuries?
Page 12, Line 436: What do you mean by 'exhibited significant deficits'? Please explain.
It was unclear what the rationale was by comparing two quite different sports in this way. I am not sure what you have achieved by doing this in terms of adding to existing literature.
Comments on the Quality of English Language
There are only one or two minor corrections that need to be made, with respect to English Language. Generally very good.
Round 2
Reviewer 1 Report
Comments and Suggestions for Authors
I would like to thank the authors for accepting the suggestions and implementing the necessary revisions. The manuscript is now significantly improved and presents a clearer and more valuable contribution to the field.